# Assessment of Land Reclamation Benefits in Mining Areas Using Fuzzy Comprehensive Evaluation

**Xueyi Yu** [1,2], **Chi Mu** [1,2,3,]*[ID] **and Dongdong Zhang** [1,2,3]

[1] School of Energy Engineering, Xi'an University of Science and Technology, Xi'an 710054, China; yuxy@xust.edu.cn (X.Y.); xust_zhangdd0629@163.com (D.Z.)

[2] Key Laboratory of Western Mine Exploitation and Hazard Prevention with the Ministry of Education, Xi'an University of Science and Technology, Xi'an 710054, China

[3] Shaanxi Provincial Land Engineering Construction Group, Xi'an 710054, China

[*] Correspondence: muc@stu.xust.edu.cn; Tel.: +86-135-7224-7696

**Abstract:** Land reclamation plays a vital role in the ecological improvement and economic development of mining regions. This study aims to conduct a preliminary discussion on the evaluation content, evaluation methods, and evaluation indicators of land reclamation benefits in mining areas. Using fuzzy comprehensive evaluation (FCE) method, land reclamation was assessed. After compiling a model of the land reclamation influencing factors, an evaluation index of land reclamation benefit in the mining area was constructed using the land reclamation monitoring data for the northern part of the mining area over the last decade. In addition, an expert scoring method and a traditional evaluation model were used to estimate the comprehensive benefits of land reclamation at Hanjiawan coal mine in Shendong mining area. Land reclamation markedly improved the land type within the mining region and decreased the amount of damaged land, including subsided and occupied land. Moreover, land reclamation improved the available land area such as agricultural and construction land. The proposed model obtained an overall 63% increase in the land reclamation area. Different degrees of ecological, economic, and social benefits of Hanjiawan coal mine were observed; however, the ecological benefits were the most significant, with a growth rate of 56%. Based on the evaluation criteria, all benefits of the mining area after reclamation were good. Over time, land reclamation will offer greater comprehensive benefits to the mining area. Furthermore, this method can be used for precise evaluation of comprehensive benefits after land reclamation, and the assessment results will provide a reference basis for sustainable development of the mining area.

**Keywords:** Shendong mining area; land reclamation; fuzzy comprehensive evaluation (FCE); evaluation index; sustainable development

## 1. Introduction

Shendong mining area is encountering a series of environmental problems because of the large-scale exploitation of coal resources. Environmental issues such as surface movement and deformation caused by goaf collapse have altered the original stratum structure and accounted for serious geological disasters, such as landslides and debris flow in some mining areas. The extensive mining of mineral resources affects the native water resources, and soil imbalance decreases the surface water level, loosens soil, and increases land desertification [1–3]. Meanwhile, excessive mining destroys the original natural environment and vegetation, as well as gradually reduces the land available for agriculture, which adversely affects crop yields. All these developments negatively affect farmers' sources of income, thereby decreasing the economic sustainability of the mining area [4,5].

Recent years have witnessed an upsurge in the comprehensive research on mining land reclamation by the Chinese mining industry and research groups, making it one of the latest hot topics of research. Li et al. [6–8] based their research on geographic information system (GIS) and method of landscape ecology and measured the ecological impact of the mining area with an indices system constructed with optimized landscape pattern, assigning the weight for each index using the analytic hierarchy process (AHP) method. Xu et al. [9,10] investigated an assessment model of the land ecological quality (LEQ). Taking the coal mining area in Jiawang as the case study, Xu et al. assessed the LEQ in 2001 and 2010, and analyzed its changes based on remote sensing (RS) and GIS technology. According to the ecological restoration, Bian et al. [11–14] highlighted the crucial role of natural restoration and artificial intervention on land reclamation. Li et al. [15,16] focused on the land reclamation of Zhangji mine area by applying space and time correlation method; besides, they systematically examined the physical and chemical properties of reclamation soil in Zhangji reclamation area. Furthermore, Yin [17] adopted the hybrid VIKOR method to develop an evaluation model of land reclamation in an open-pit mine to evade the impact of subjective and objective factors.

Iranian scholar Abaidoo et al. [18] used artificial neural network (ANN) to monitor reclamation activities in a small-scale mining area and developed the normalized difference vegetation index (NDVI) to estimate where actual change had occurred and to what extent it had occurred. Yu et al. [19,20] investigated the mechanisms of coal mining and reclamation that affect soil properties (i.e., physical, chemical and biological) and elucidated soil development in reclamation. Based on the collected data of each land, Wu et al. [21,22] evaluated the integrated fertility index (IFI) of every land to denote the productivity levels of reclamation lands in mining districts. Ghanaian scholar Bakhtavar et al. [23] proposed a new method to assess and optimize the mining reclamation strategy by combining an intelligent multi-objective fuzzy cognitive map with fuzzy analysis network process. Wang et al. [24,25] assessed the Antaibao opencast coal mine of the Loess area in China to obtain data regarding soil macropores using a high-resolution, nondestructive computed tomography (CT) technique. American scholar Swab et al. [26] combined several species in standard reclamation mixes with prairie species native to North America to create a higher diversity planting on three mine sites in southeastern Ohio.

However, most of studies mentioned above were based on the land reclamation and ecological environment construction in mining areas. Thus, the comprehensive benefit research on land reclamation in the mining area warrants further investigation. Hence, it is essential to establish a land reclamation benefit index system and conduct comprehensive research.

Many types of evaluation methods are available. Based on the evaluation results of land reclamation in mining areas, experts and scholars commonly use the following evaluation methods: analytic hierarchy process (AHP); fuzzy comprehensive evaluation (FCE); principal component analysis (PCA); fault tree analysis (FTA); and logical framework approach (LFA). Notably, all evaluation methods have various differences. Table 1 compares the common evaluation methods.

Land reclamation benefits in mining areas can be examined by ordering and optimizing multiple plans. Each plan warrants comprehensive evaluation, ranking, and optimizing the plans based on certain evaluation criteria. Furthermore, based on the comparative analysis of the advantages and disadvantages of the evaluation methods mentioned above, this study uses the FCE method as the evaluation method. The comprehensive development of Shendong mining area is holistically considered in this study to optimize the land use rate and enhance the ecological environment of the mining area. The feasibility of land reclamation in the mining area was assessed by focusing on the factors influencing the land reclamation quality, such as the ecological landscape effect, agricultural land reclamation area, and satisfaction level of the residents. In addition, an evaluation model of land reclamation in the mining area was constructed through a consistency test of a judgment matrix. Moreover, the ecological, economic, and social benefits before and after land reclamation in the mining area were simultaneously evaluated by an expert scoring method based on the evaluation criteria. Overall, the resulting benefit scores provided a scientific basis for comprehensively evaluating the effects of land reclamation in Shendong mining area.

**Table 1.** Comparison of evaluation methods.

| Method | Feature | Advantage | Disadvantage |
|---|---|---|---|
| Analytic hierarchy process (AHP) | Combination of qualitative and quantitative analysis. | Systematically treat research objects as a system and make decisions in a comprehensive way of thinking. | Cannot provide new options for decision-making. |
| Fuzzy comprehensive evaluation (FCE) | Combination of AHP and fuzzy mathematics. | The problem of quantification of a large number of uncertain factors in the evaluation is well solved. | It is more subjective when determining fuzzy relation matrix and factor weight. |
| Principal component analysis (PCA) | Determine the number of principal components through appropriate mathematical changes. | Can eliminate the mutual influence between evaluation indicators and reduce calculation workload. | The raw data requirements are complete and the evaluation process is not flexible. |
| Fault tree analysis (FTA) | Create a list of key events for different importance measurements. | Use graphics to clearly show the process and results of the evaluation object. | It is no way to accurately calculate system failure probability and reliability. |
| Logical framework approach (LFA) | Use simple block diagrams to analyze complex relationships. | Clarify the cause and effect in the project. | Greatly affected by important assumptions. |

## 2. Requirements of Land Reclamation

From the general law of the global mining area development, the optimization and transformation of the industrial structure are the inevitable choice for the mining area to attain sustainable development. Mining areas provide raw materials and energy for economic construction and people's production; however, the exploitation of mineral resources inevitably damages ecological resources such as land resources, water resources, and biological resources [27]. Based on the sustainable use of reclaimed land, the mineral resources industry structure adjustment and optimization are completed, the continuous industry or alternative industry is sought, and the enterprise is gradually transformed from resource-based to non-resource-based to ensure the sustainable development of the mining area.

Sustainable development is the ultimate goal of mine production. The sustainable development of reclaimed land in mining areas implies the process of land reuse based on land reclamation and ecological reconstruction [28]. The key to the sustainable development of mining areas lies in the coordination with social, ecological, and economic benefits. This study examined the sustainable development of the mining area in terms of changes in comprehensive benefits before and after land reclamation. Figure 1 shows the technical route.

The mining area provides raw materials and energy for economic construction and people's production, but the development of mineral resources inevitably causes damage to land resources. The reuse of abandoned mining areas directly restricts the sustainable development of the mining area [29]. If extreme damage occurs to surface water and soil resources and high-intensity mining of mineral resources, artificial induction should be used to actively increase capital investment, carry out land reclamation and continuation, and replace industrial development, and build a stable mining area complex ecosystem to enable the mining area to maintain a high-level sustainable development [30].

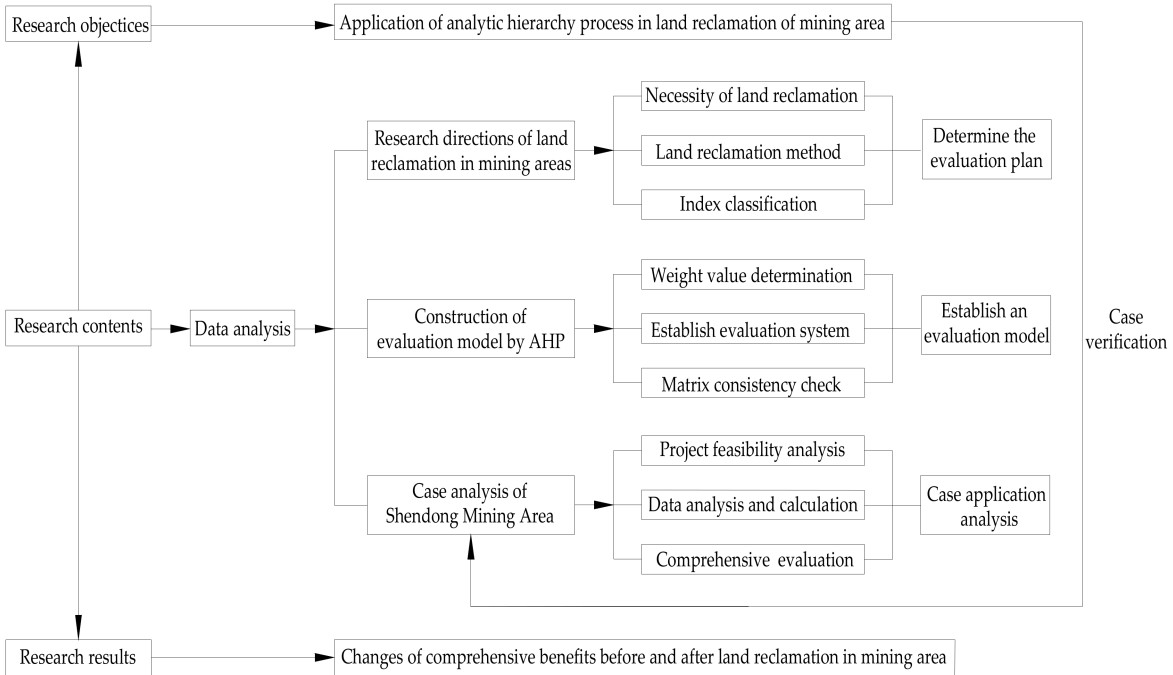

**Figure 1.** The technical route of land reclamation benefit evaluation in the mining area.

Of note, the sustainable development of the mining area should meet three conditions. Firstly, sustainable development of ecological benefits. Efforts should be made to coordinate the resources of the mining area with the environmental carrying capacity and ensure that the ecological environment of the mining area remains healthy. Secondly, sustainable economic development. The mining area should pay attention to saving resources and reducing pollutant emissions while paying attention to the rate of economic growth. Thirdly, sustainable development of social benefits. The purpose is to enhance the living environment of the residents of the mining area and adapt to social progress. Comprehensive benefit evaluation is essential to the healthy development of the mining area.

## 3. Feasibility Analysis of Land Reclamation in the Mining Area

### 3.1. Overview

This section provides the overall overview of the study area. Hanjiawan coal mine is located at the northern end of Shendong mining area. The north–south exploration area is 6.5–10.8 km long, and the exploration area is 46.52 km$^2$; it is located in Yulin City, Shaanxi Province. The area belongs to the low mountain hills of the Loess Plateau in northern Shaanxi (Figure 2). The typical continental monsoon climate is dry and cold with relatively low temperatures. The annual temperature difference is higher than the daily temperature difference, and the average annual temperature is 3.6 °C–15.7 °C. Furthermore, the annual precipitation distribution is extremely uneven, and the intensity of the heavy rain is 361.5–635.1 mm.

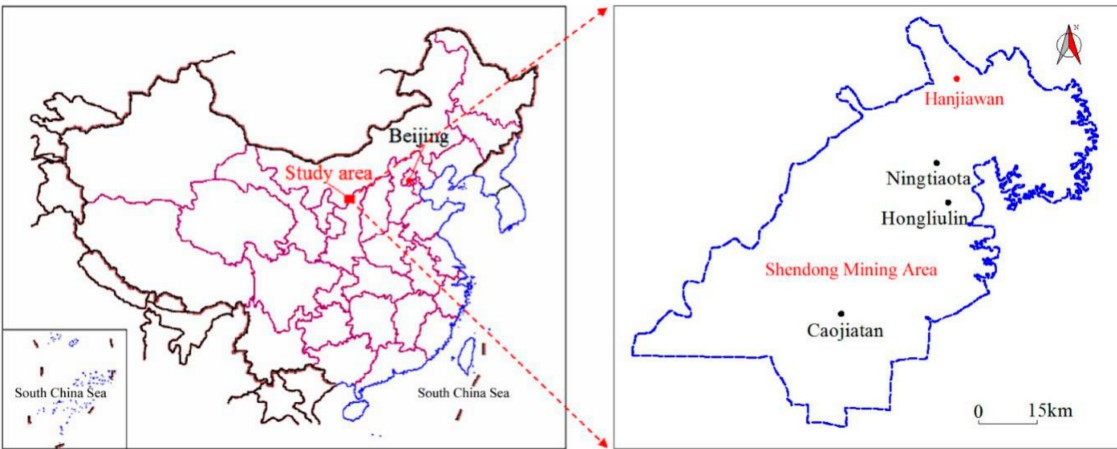

**Figure 2.** Location of the study area.

### 3.2. Influence of Mining on Land Resources

Mine subsidence occurs because of the collapse of underground mineral resources and generally manifests as collapse pits, ground fissures, and landslides. The large-scale mining of mineral resources in Shendong mining area has not only destroyed the natural and human environment of the original land, but also caused the loss of surface water and soil, rendering the resulting surface barren and prone to desertification. Moreover, the large-scale mining minerals caused landslides, earthquakes, and other geological disasters. For example, in the Hanjiawan Mine of Shendong mining area, serious surface deformation has been caused by mining, resulting in surface cracks as wide as 2 m and as deep as 15–20 m, as shown in Figure 3.

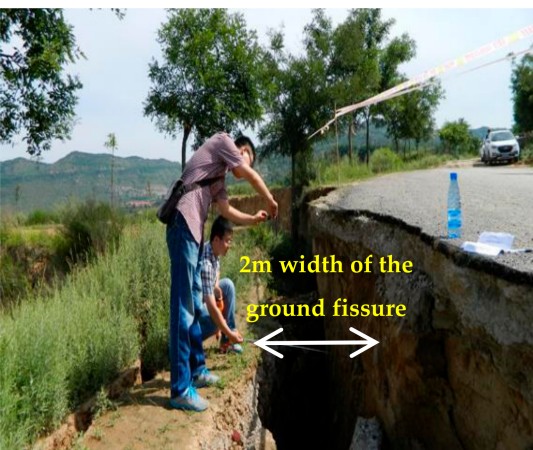

**Figure 3.** Surface collapse pit.

Large-scale mining in Northern Shaanxi has caused surface deformation, affecting the surface soil structure. Mining regions usually possess a loose soil structure, low soil moisture, decreased soil erosion resistance, and lower land productivity. Mining activities in these regions account for the development of ground fissures, resulting in the penetration of harmful substances into underground soil and water. Moreover, soil erosion and eventual desertification have been observed in the northern mining regions due to large-scale mining, accompanied by a decline in the water level, surface subsidence, ground fissure, damage to vegetation, soil erosion, pollution and other problems (Figure 4).

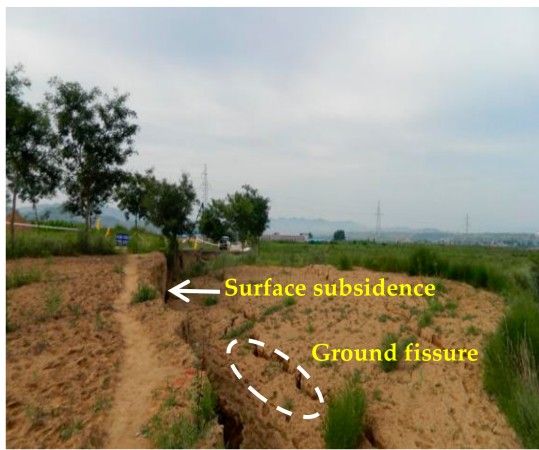

**Figure 4.** Surface desertification.

### 3.3. Evaluation of Land Destruction in the Mining Area

The surface damage of Shendong mining area was graded and divided into three categories: slight damage, moderate damage, and severe damage. In the absence of widely-accepted specific rating criteria, the actual investigation of mine damage and the research status of land reclamation in mining areas were executed according to the following criteria:

### 3.3.1. Evaluation Standard of the Subsided Land

In lieu of an accurate classification of land collapse damage grades in mining areas, this paper evaluated the practical experience data of related disciplines by using the leading factor method. The evaluation was based on the real-time investigation of similar engineering land damage in the Shaanxi Province. Table 2 lists the factors and classification of land collapse damage in mining areas.

**Table 2.** Evaluation factors and grade standard for estimation of the degree of land collapse.

| Evaluation Grade | Evaluation Factor | | |
| --- | --- | --- | --- |
| | Collapse Depth (m) | Subsidence Area (m$^2$) | Thickness of Surface Soil (m) |
| Mild damage | <0.5 | <0.5 | <0.2 |
| Moderate damage | 0.5–2.0 | 0.5–1.0 | 0.2–0.5 |
| Severe damage | >2.0 | >1.0 | >0.5 |

### 3.3.2. Land Occupation Evaluation Criteria

Based on the real-time investigation of the occupied land in the Shaanxi Province mining area and referring to the global research literature, we used a fuzzy factor synthesis to classify the occupied land in the mining area, as shown in Table 3.

**Table 3.** Compression factors and grading standards for the discarded land.

| Evaluation Grade | Evaluation Factor | | |
| --- | --- | --- | --- |
| | Compression Area (m$^2$) | Earthwork Height (m) | Slope Angle (°) |
| Mild damage | <1 | <1 | <15 |
| Moderate damage | 1–5 | 1–3 | 15–30 |
| Severe damage | >5 | >3 | >30 |

### 3.4. Current Status of Land Reclamation in Mining Area

The goaf area formed by Shendong mining area has reached 42 km$^2$. The damaged land area is nearly 56.21 km$^2$. The land reclamation area of the mining area is 9.82 km$^2$, with the reclamation rate

of only 10.2%. We learned from the local government that by 2020, the land reclamation rate of the mining area will increase by 60%, and the reclamation area will reach 1810 km$^2$. The primary types of land reclamation in Shendong mining area include the following: agricultural land, construction land, mining area land, forestry land and other land. Table 4 shows the land reclamation area of Shendong mining area.

**Table 4.** Land reclamation in Shendong mining area from 2009 to 2018 (km$^2$).

| Factor | Agricultural Land | Construction Land | Mining Area Land | Forestry Land | Other Land |
|---|---|---|---|---|---|
| Subsided land | 3025 | 1109 | 1253 | 1362 | 351 |
| Waste land | 691 | — | 237 | — | 117 |
| Fouling land | 811 | 137 | 162 | 51 | — |
| Saline-alkali land | 1056 | — | — | 180 | 191 |
| Total | 5583 | 1246 | 1652 | 1593 | 659 |

### 3.5. Suitability of Land Reclamation in the Mining Area

Shendong mining area has the characteristics of thick loose layer on the surface, weak soil water storage capacity, and poor natural conditions of the land. After the land destruction, recovery is challenging because of multiple factors. The land reclamation impact factor is high through digging and filling treatment methods. The land reclamation in Shendong mining area is assessed by the minimum impact factor. The limiting factors and grading indicators are determined by a single factor (minimum limiting factor), and the single factor is used as the decision layer of the evaluation system, which exerts a major impact on the reclamation factors affecting the mining area. Considering the main evaluation factors of Shendong mining area, the grades of the factors affecting land reclamation are obtained (Table 5).

**Table 5.** Grade standard for influencing factors of land reclamation in the mining area.

| Influencing Factors | | Cultivated Land | Forest Land | Grassland |
|---|---|---|---|---|
| Surface composition | Sand | 1 | 1 | 1 |
| | Rock and soil | 2 | 3 | 2 |
| | Rock | 3 | 3 | 3 |
| Thickness of surface soil/mm | >500 | 1 | 1 | No effect |
| | 200–500 | 2 | 2 | No effect |
| | <200 | 3 | 2 | No effect |
| Land slope/(°) | <10 | 1 | 1 | 1 |
| | 10–30 | 2 | 1 | 3 |
| | >30 | 3 | 3 | 3 |

Table 5 shows that the surface sand is a vital part of forest land, cultivated land and grassland; higher thickness of the topsoil contributes to the reclamation of the mining area, and the smaller the surface slope, the smaller the impact on land destruction. Thus, based on the land reclamation target of the mining area, along with the regional environmental conditions and economic development characteristics, the scientific and reasonable adjustment of the minimum factors affecting the land reclamation in the mining area will increase the nutrients of the surface soil and make the forest land, cultivated land and grassland better adapt to surface materials, improving the land reclamation effect of the mining area.

## 4. Establishment of Comprehensive Evaluation Model

The comprehensive benefit evaluation of land reclamation in Shendong mining area is based on the evaluation model structure. Figure 5 shows the comprehensive evaluation analysis of land reclamation in the mining area.

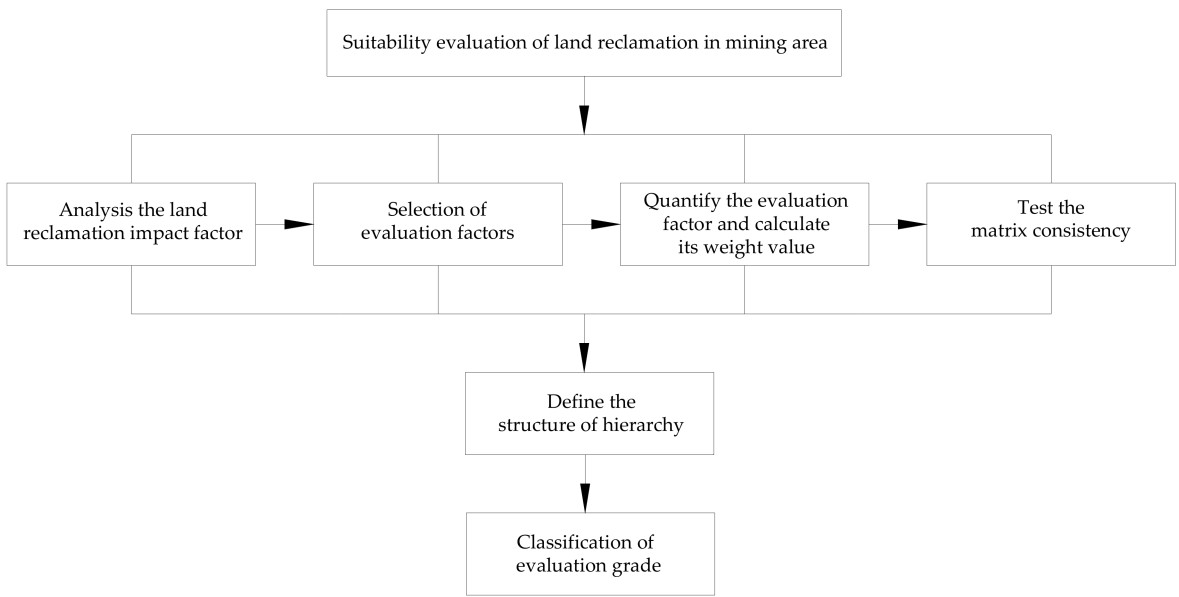

**Figure 5.** Analysis of comprehensive evaluation.

### 4.1. Comprehensive Evaluation Model

FCE is a comprehensive evaluation of something using fuzzy mathematical tools. The basic principle of FCE is as follows: first determine the factor set $U = \{u_1, u_2, u_3, \ldots, u_n\}$ and decision set $V = \{v_1, v_2, v_3, \ldots, v_m\}$ of the object being judged. Here, $u_i$ denotes each single index and $v_i$ denotes the rating level of $u_i$. After fuzzy transformation, a fuzzy evaluation matrix $R$ is obtained, and then the weight $A$ of each factor is determined. Finally, the fuzzy evaluation matrix and factor weight vector set are fuzzy-calculated and normalized to obtain FCE result set B. $B = A \cdot R$. Then, (*U, V, A, R*) constitutes a comprehensive evaluation model [31].

1. Determine factor set $U = \{u_1, u_2, u_3, \ldots, u_n\}$ and decision set $V = \{v_1, v_2, v_3, \ldots, v_m\}$.

2. Establish fuzzy relation matrix *R*.

Constructing the hierarchical fuzzy subset, we must quantify the evaluated things one by one from each factor, that is, ascertain the degree of membership of the evaluated things to the hierarchical fuzzy subset from a single factor to obtain the fuzzy relationship matrix:

$$R = \begin{bmatrix} r_{11} & r_{12} & K & r_{1m} \\ r_{21} & r_{22} & K & r_{2m} \\ M & M & M & M \\ r_{n1} & r_{n2} & K & r_{nm} \end{bmatrix} \tag{1}$$

The element $r_{ij}$ in the *i*-th row and the *j*-th column in the matrix $R$ denotes the degree of membership of the $v_j$ fuzzy subset from the factor $u_i$, so $R$ also becomes the degree of membership matrix.

3. Determine the weight vector of the evaluation factors *A*.

In FCE, AHP is used to evaluate the weight vector of evaluation factors: $A = \{a_1, a_2, a_3, \ldots, a_n\}$, $0 \leqq a_i \leqq 1$, $\sum_{i=1}^{n} a_i = 1$. AHP determines the relative significance between factors to determine the weight coefficients and normalizes them before synthesis [32].

4. Establishment of FCE matrix *B*.

The principle of maximum membership is the most commonly used method in practice; however, its disadvantage is that it loses considerable information in some cases and even obtains unreasonable evaluation results. Thus, in this study, the FCE method used a "weighted average" model, where A and R were synthesized by using a suitable operator to attain an FCE result vector B of each evaluated object [33], which is:

$$B = A \cdot R = (a_1, a_2, a_3, \ldots, a_n) \cdot \begin{bmatrix} r_{11} & r_{12} & K & r_{1m} \\ r_{21} & r_{22} & K & r_{2m} \\ M & M & M & M \\ r_{n1} & r_{n2} & K & r_{nm} \end{bmatrix} = (b_1, b_2, b_3, \ldots, b_n) \quad (2)$$

Among them, $b_i$ denotes the degree of membership of the rated thing to the fuzzy subset of $v_i$.

### 4.2. Land Reclamation Method

#### 4.2.1. Terraced Land Reclamation

The slope of the additional land generated by coal mining subsidence is usually relatively small, within about 2°. When the ground can be cultivated through land levelling or unevenness, it can be repaired along the contour line when the slope of the ground surface is 2°–6°; it is also terraced and inclined slightly inward. When land is used, it can be laid out between agriculture and forestry [34]. Besides, contour cultivation can be used for soil and water conservation; such land reclamation is terraced reclamation. For mining areas located in hilly areas or in middle coal seams, the damage of cultivated land is characterized by unevenness or even step-like landforms. Furthermore, terraced reclamation is suitable for subsidence basins located in hilly and mountainous areas or for large surface slopes after mining subsidence in low- and medium-water table mining areas.

#### 4.2.2. Artificial Afforestation

After mining, the land reclamation technology for planting forests can rapidly form green vegetation on the land, protect the soil from soil erosion, increase the land fertility, and improve the regional ecological conditions. The use of afforestation technology for land reclamation in mining areas can typically obtain better results; the key to this reclamation method is the selection of tree species [35]. In various countries, tree species selection has been the focus of reclamation research. American scholar Cruickshank et al. [36] comprehensively studied this area and accrued rich production experience and scientific research results. The selected plants should have bio-ecological characteristics such as resistance to pollution, fast growth, and good soil and water conservation.

#### 4.2.3. Multiple Methods Combined

Typically, land reclamation is implemented through engineering reclamation to eliminate accumulated water, droughts, and floods. Biological reclamation methods are used to enhance soil quality and regulate environmental pollution, and soil pollution due to mining; and ecological reclamation methods are used to improve reclamation of the local ecological environment of the land, not only enabling the land to be reclaimed but also realizing the real comprehensive benefits of the mining area [37].

FCE is the combination of AHP and fuzzy mathematics. Of note, AHP effectively solves multilevel and multi-objective decision-making problems in large systems and has the characteristics of high logic, flexibility, and simplicity. The land reclamation project in the mining area involves various factors and contains complicated links. During reclamation, there could be many methods to choose from. We can sort out several representative methods, apply AHP for selection analysis, and obtain the best reclamation effect with as few reclamation methods as possible, as well as obtain good ecological, economic, and social benefits.

### 4.3. Determining the Grade and Evaluation System of Land Reclamation in Mining Area

This study discusses the appropriate land reclamation research results, based on the land reclamation status in specific mining areas in China. We selected five evaluation criteria, and defined the commentary set as "Excellent", "Good", "Average", "Poor", and "Very poor". Table 6 summarizes the comprehensive benefit evaluation standards for land reclamation in the mining area.

**Table 6.** Grade standard for evaluating land reclamation benefit in the mining area.

| Grade | 1 | 2 | 3 | 4 | 5 |
|---|---|---|---|---|---|
| Benefit evaluation standard | Excellent | Good | Average | Poor | Very poor |
| Assignment level (AL) | 80–100 | 60–80 | 40–60 | 20–40 | <20 |

By systematically analyzing relevant theories of land reclamation benefit evaluation, we analyzed and screened the evaluation index of land reclamation benefits in mining areas. Relevant research results, along with expert opinions and the selection principle of land reclamation benefit evaluation index, were also used in the analysis and screening. Starting from ecological environment protection and sustainable economic development, and combining this with the real production of the mining area, we selected the impact indicators suitable for land reclamation area. The ecological, economic, and social benefits were used as the criteria layer [38–40]. Overall, 15 variables were used as the index layer to establish a land reclamation evaluation system for the mining area; Table 7 shows the calculation method of each index.

**Table 7.** Evaluation index and calculation method of land reclamation in mining area.

| Target Layer | Criteria Layer | Index Layer | The Origin of the Data |
|---|---|---|---|
| Overall benefits A | Ecological benefits B1 | Land reclamation rate in the mining area C1<br>Green vegetation coverage C2<br>Irrigation guarantee rate C3<br>Biodiversity in the mining area C4<br>Land reclamation quality C5<br>Ecological landscape effect C6 | Government databases<br>Digital maps<br>Government databases<br>Academic journal databases<br>Institutional investigation<br>Digital maps |
| | Economic benefits B2 | Land use efficiency in mining area C7<br>Input-output ratio C8<br>Per capita income of residents C9<br>Mechanized operating rate C10<br>Project budget costs C11 | Government databases<br>Institutional investigation<br>Network survey<br>Institutional investigation<br>Experimental databases |
| | Social Benefits B3 | Residents' satisfaction C12 | Public survey |
| | | Basic equipment matching rate C13<br>Grain yield of reclaimed farmland C14<br>Guaranteed employment rate C15 | Institutional investigation<br>Government databases<br>Network survey |

### 4.4. Establishing Land Reclamation Evaluation Model for the Mining Area

The research objective was to evaluate land reclamation benefits by analyzing the land reclamation quality, ecological landscape effects, agricultural land area, and resident income level in the study region [41,42]. We used an analytical hierarchy process to comprehensively evaluate the land reclamation benefit of the mining area. Based on different evaluation index attributes, the land reclamation evaluation was divided into four layers (Target layer, Criteria layer, Indicator layer, and Solution layer), to establish a comprehensive benefit evaluation model, as shown in Figure 6.

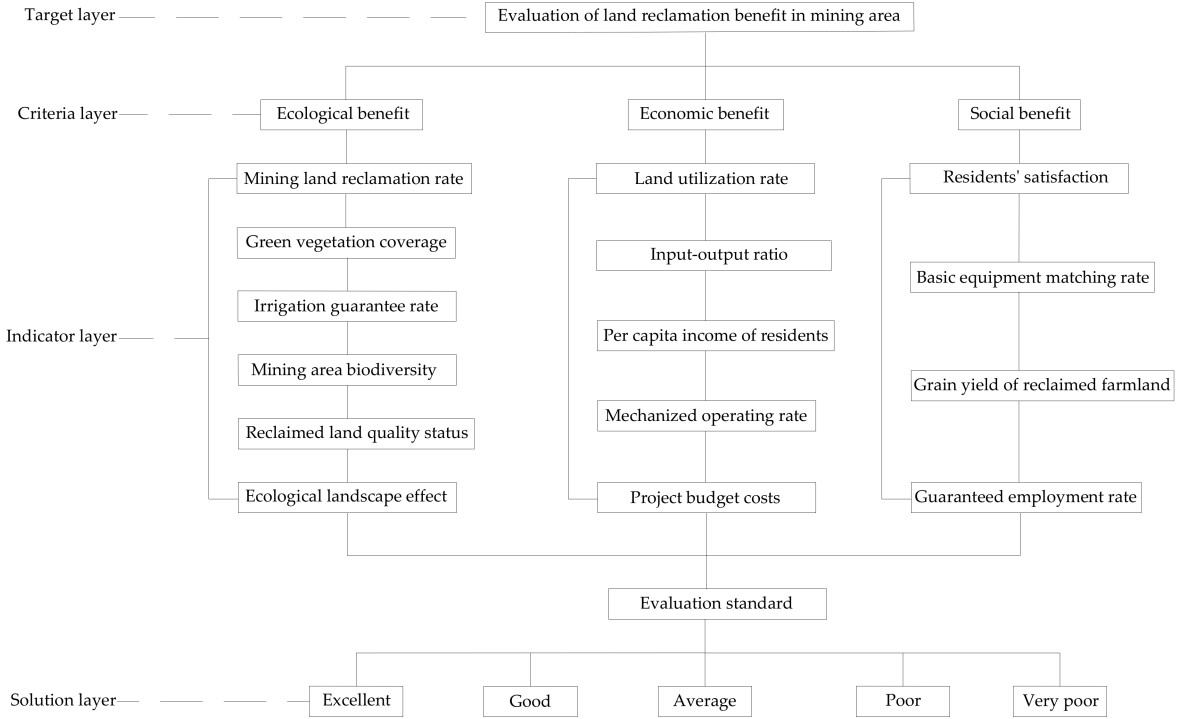

**Figure 6.** Evaluation model of the land reclamation benefit in the mining area.

*4.5. Consistency Check of Judgment Matrix*

Based on the AHP, we calculated the weights of the variables of each index layer. In addition, an expert scoring method was used to compare the variables of each structural layer per the basic principles of the evaluation system. Using the expert scoring method, we obtained the discriminant matrix and calculated the maximum eigenvalue of the matrix. The maximum eigenvalue and eigenvector of the matrix were calculated from Equations (3) and (4).

$$l_j = \sum_{j=1}^{m} r_{ij} - 0.5, j = 1, 2, 3, \mathrm{K}, \, m \tag{3}$$

$$\sum_{i} r_{ij} = \frac{m(m-1)}{2} \tag{4}$$

where $l_j$ is the sum of the elements of each row of the matrix, and $\sum_{i} r_{ij}$ is the sum of matrix elements. The weight value of each evaluation factor was calculated from Equation (5).

$$w_i = \frac{l_i}{\sum_{i} l_i} = \frac{2l_i}{m(m-1)} \tag{5}$$

where $w_i$ is the weight value of the evaluation matrix factor. Based on the weight value of the judgment matrix, we introduced the scale values of nine different levels. The average value of the difference between the maximum eigenvalue and the matrix dimension was used as an index to measure the consistency of the judgment matrix, as expressed by Equation (6).

$$CI = \frac{\lambda_{\max} - n}{n - 1} \tag{6}$$

where $\lambda_{\max}$ denotes the maximum eigenvalue, and $n$ is the matrix dimension.

To determine whether different class matrices are consistent, we introduced the randomness index (RI) of the judgment matrix; Table 8 presents the value of RI for the judgment matrix of $n$ = 1–14.

**Table 8.** Judgment matrix randomness index.

| Order | 1 | 2 | 3 | 4 | 5 | 6 | 7 | 8 | 9 | 10 | 11 | 12 | 13 | 14 |
|---|---|---|---|---|---|---|---|---|---|---|---|---|---|---|
| **RI** | 0 | 0 | 0.52 | 0.89 | 1.12 | 1.24 | 1.36 | 1.41 | 1.46 | 1.49 | 1.52 | 1.54 | 1.56 | 1.58 |

When the order of the matrix was >1, the complete consistency (CI) of the judgment matrix and the RI ratio of the same-order average random number consistency index, and random consistency ratio were recorded as satisfactory consistency (CR), where $CR = \frac{CI}{RI} < 0.1$. The calculation results of the hierarchy satisfied the consistency of the judgment matrix, otherwise the index value of the matrix should be adjusted to ensure consistency.

## 5. Benefit Evaluation of Land Reclamation in Shendong Mining Area

*5.1. Evaluation Index System of Land Reclamation in the Mining Area*

We evaluated the comprehensive benefit of the land reclamation in Hanjiawan coal mine of Shendong mining area. Combined with the geological environment of the mine, the factors influencing the benefits in the mining region were sorted and screened. In addition, we compared the function factors of each index and established an evaluation system of land reclamation in Hanjiawan coal mine. Based on the fuzzy evaluation method, the ecological, social, and economic benefits were selected as the comprehensive benefit index layers. The weight of the criterion layer was estimated to be A = (0.5102, 0.3151, 0.1788), and the consistency test was passed since CR = 0.0812 (i.e., <0.1). Hence, it was possible to confirm that the weight of the criterion level was consistent [43–46].

The direct selection of experts is based on multiple factors. We quantified the standard layer based on basic data and expert scores (expert group of industry experts). Table 9 presents the evaluation results before and after land reclamation based on the expert judgment.

**Table 9.** The hierarchical weights and expert scoring results.

| Target Layer | Criteria Layer | Indicator Layer | Experts' Evaluation Results Before and After Reclamation | | | | |
|---|---|---|---|---|---|---|---|
| | | | **Excellent** | **Good** | **Average** | **Poor** | **Very Poor** |
| A | B1 (0.5102) | C1 (0.2031) | 0.0<br>0.1 | 0.2<br>0.2 | 0.1<br>0.4 | 0.4<br>0.3 | 0.5<br>0.1 |
| | | C2 (0.1343) | 0.0<br>0.4 | 0.5<br>0.1 | 0.7<br>0.1 | 0.1<br>0.2 | 0.3<br>0.2 |
| | | C3 (0.2190) | 0.0<br>0.1 | 0.3<br>0.7 | 0.2<br>0.2 | 0.2<br>0.6 | 0.1<br>0.6 |
| | | C4 (0.1215) | 0.0<br>0.6 | 0.1<br>0.5 | 0.3<br>0.5 | 0.1<br>0.2 | 0.2<br>0.3 |
| | | C5 (0.2190) | 0.0<br>0.8 | 0.7<br>0.2 | 0.1<br>0.6 | 0.5<br>0.5 | 0.2<br>0.8 |
| | | C6 (0.1031) | 0.0<br>0.3 | 0.1<br>0.7 | 0.1<br>0.3 | 0.5<br>0.4 | 0.8<br>0.2 |

**Table 9.** *Cont.*

| Target Layer | Criteria Layer | Indicator Layer | Experts' Evaluation Results Before and After Reclamation | | | | |
|---|---|---|---|---|---|---|---|
| | | | **Excellent** | **Good** | **Average** | **Poor** | **Very Poor** |
| A | B2 (0.3151) | C7 (0.2103) | 0.0<br>0.5 | 0.1<br>0.1 | 0.2<br>0.1 | 0.0<br>0.1 | 0.2<br>0.2 |
| | | C8 (0.1359) | 0.0<br>0.7 | 0.0<br>0.0 | 0.1<br>0.4 | 0.2<br>0.2 | 0.1<br>0.7 |
| | | C9 (0.2058) | 0.0<br>0.3 | 0.6<br>0.3 | 0.7<br>0.6 | 0.1<br>0.6 | 0.0<br>0.1 |
| | | C10 (0.2510) | 0.0<br>0.0 | 0.2<br>0.5 | 0.5<br>0.8 | 0.6<br>0.3 | 0.8<br>0.2 |
| | | C11 (0.1970) | 0.0<br>0.1 | 0.1<br>0.6 | 0.3<br>0.0 | 0.1<br>0.1 | 0.4<br>0.3 |
| | B3 (0.1788) | C12 (0.4105) | 0.0<br>0.3 | 0.4<br>0.2 | 0.3<br>0.2 | 0.0<br>0.1 | 0.1<br>0.1 |
| | | C13 (0.2551) | 0.0<br>0.1 | 0.1<br>0.0 | 0.7<br>0.3 | 0.1<br>0.4 | 0.3<br>0.4 |
| | | C14 (0.1183) | 0.0<br>0.5 | 0.6<br>0.6 | 0.1<br>0.4 | 0.8<br>0.2 | 0.2<br>0.1 |
| | | C15 (0.2161) | 0.0<br>0.5 | 0.4<br>0.7 | 0.5<br>0.7 | 0.6<br>0.2 | 0.0<br>0.2 |

## 5.2. Evaluation of Land Reclamation Efficiency in the Mining Area

In this study, the membership degree of each index was determined on the basis of the benefit evaluation of reclamation, combined with literature data. The five-level standard of the situation before and after the re-recovery of each benefit evaluation index in Shendong mining area was evaluated using fuzzy statistical analysis to obtain various indicators. The membership degree and the fuzzy evaluation matrix before and after land reclamation were established as shown in the matrices below.

Pre-reclamation relationship matrix Post-reclamation relationship matrix

$$R_{pre-recovery\ ecology} = \begin{Bmatrix} 0.0 & 0.2 & 0.1 & 0.4 & 0.5 \\ 0.0 & 0.5 & 0.7 & 0.1 & 0.3 \\ 0.0 & 0.3 & 0.2 & 0.2 & 0.1 \\ 0.0 & 0.1 & 0.3 & 0.1 & 0.2 \\ 0.0 & 0.7 & 0.1 & 0.5 & 0.2 \\ 0.0 & 0.1 & 0.1 & 0.5 & 0.8 \end{Bmatrix}$$

$$R_{post-recovery\ ecology} = \begin{Bmatrix} 0.1 & 0.2 & 0.4 & 0.3 & 0.1 \\ 0.4 & 0.1 & 0.1 & 0.2 & 0.2 \\ 0.1 & 0.7 & 0.2 & 0.6 & 0.6 \\ 0.6 & 0.5 & 0.5 & 0.2 & 0.3 \\ 0.8 & 0.2 & 0.6 & 0.5 & 0.8 \\ 0.3 & 0.7 & 0.3 & 0.4 & 0.2 \end{Bmatrix}$$

$$R_{pre-recovery\ economy} = \begin{Bmatrix} 0.0 & 0.1 & 0.2 & 0.0 & 0.2 \\ 0.0 & 0.0 & 0.1 & 0.2 & 0.1 \\ 0.0 & 0.6 & 0.7 & 0.1 & 0.0 \\ 0.0 & 0.2 & 0.5 & 0.6 & 0.8 \\ 0.0 & 0.1 & 0.3 & 0.1 & 0.4 \end{Bmatrix}$$

$$R_{post-recovery\ economy} = \left\{ \begin{array}{ccccc} 0.5 & 0.1 & 0.1 & 0.1 & 0.2 \\ 0.7 & 0.0 & 0.4 & 0.2 & 0.7 \\ 0.3 & 0.3 & 0.6 & 0.6 & 0.1 \\ 0.0 & 0.5 & 0.8 & 0.3 & 0.2 \\ 0.1 & 0.6 & 0.0 & 0.1 & 0.3 \end{array} \right\}$$

$$R_{pre-recovery\ society} = \left\{ \begin{array}{ccccc} 0.0 & 0.4 & 0.3 & 0.0 & 0.1 \\ 0.0 & 0.1 & 0.7 & 0.1 & 0.3 \\ 0.0 & 0.6 & 0.1 & 0.8 & 0.2 \\ 0.0 & 0.4 & 0.5 & 0.6 & 0.0 \end{array} \right\}$$

$$R_{post-recovery\ society} = \left\{ \begin{array}{ccccc} 0.3 & 0.2 & 0.2 & 0.1 & 0.1 \\ 0.1 & 0.0 & 0.3 & 0.4 & 0.4 \\ 0.5 & 0.6 & 0.4 & 0.2 & 0.1 \\ 0.5 & 0.7 & 0.7 & 0.2 & 0.2 \end{array} \right\}$$

We calculated the discriminant matrix of ecological, economic, social, and comprehensive benefits before and after land reclamation by combining the present land reclamation situation and the evaluation matrix of benefits before and after reclamation. In addition, the discriminant matrix was estimated using the fuzzy evaluation discriminant equation $B = A \cdot R$, and the calculated results were normalized.

(1) Ecological benefits

$$B_{pre-recovery\ ecology} = \begin{bmatrix} 0.2031 & 0.1343 & 0.2190 & 0.1215 & 0.2190 & 0.1031 \end{bmatrix} \cdot$$
$$\left\{ \begin{array}{ccccc} 0.0 & 0.2 & 0.1 & 0.4 & 0.5 \\ 0.0 & 0.5 & 0.7 & 0.1 & 0.3 \\ 0.0 & 0.3 & 0.2 & 0.2 & 0.1 \\ 0.0 & 0.1 & 0.3 & 0.1 & 0.2 \\ 0.0 & 0.7 & 0.1 & 0.5 & 0.2 \\ 0.0 & 0.1 & 0.1 & 0.5 & 0.8 \end{array} \right\} = \begin{bmatrix} 0.0000 & 0.1343 & 0.2581 & 0.3601 & 0.0715 \end{bmatrix}$$

$$B_{post-recovery\ ecology} = \begin{bmatrix} 0.2031 & 0.1343 & 0.2190 & 0.1215 & 0.2190 & 0.1031 \end{bmatrix} \cdot$$
$$\left\{ \begin{array}{ccccc} 0.1 & 0.2 & 0.4 & 0.3 & 0.1 \\ 0.4 & 0.1 & 0.1 & 0.2 & 0.2 \\ 0.1 & 0.7 & 0.2 & 0.6 & 0.6 \\ 0.6 & 0.5 & 0.5 & 0.2 & 0.3 \\ 0.8 & 0.2 & 0.6 & 0.5 & 0.8 \\ 0.3 & 0.7 & 0.3 & 0.4 & 0.2 \end{array} \right\} = \begin{bmatrix} 0.1120 & 0.2043 & 0.3521 & 0.1613 & 0.1703 \end{bmatrix}$$

(2) Economic benefits

$$B_{pre-recovery\ economy} = \begin{bmatrix} 0.2103 & 0.1359 & 0.2058 & 0.2510 & 0.1970 \end{bmatrix} \cdot$$
$$\left\{ \begin{array}{ccccc} 0.0 & 0.1 & 0.2 & 0.0 & 0.2 \\ 0.0 & 0.0 & 0.1 & 0.2 & 0.1 \\ 0.0 & 0.6 & 0.7 & 0.1 & 0.0 \\ 0.0 & 0.2 & 0.5 & 0.6 & 0.8 \\ 0.0 & 0.1 & 0.3 & 0.1 & 0.4 \end{array} \right\} = \begin{bmatrix} 0.0000 & 0.1513 & 0.3106 & 0.4031 & 0.1350 \end{bmatrix}$$

$$B_{post-recovery\ economy} = \begin{bmatrix} 0.2103 & 0.1359 & 0.2058 & 0.2510 & 0.1970 \end{bmatrix} \cdot$$
$$\left\{ \begin{array}{ccccc} 0.5 & 0.1 & 0.1 & 0.1 & 0.2 \\ 0.7 & 0.0 & 0.4 & 0.2 & 0.7 \\ 0.3 & 0.3 & 0.6 & 0.6 & 0.1 \\ 0.0 & 0.5 & 0.8 & 0.3 & 0.2 \\ 0.1 & 0.6 & 0.0 & 0.1 & 0.3 \end{array} \right\} = \begin{bmatrix} 0.2903 & 0.1572 & 0.2011 & 0.2091 & 0.1423 \end{bmatrix}$$

(3) Social benefits

$$B_{pre-recovery\ society} = \begin{bmatrix} 0.4105 & 0.2551 & 0.1183 & 0.2161 \end{bmatrix} \cdot$$
$$\begin{Bmatrix} 0.0 & 0.4 & 0.3 & 0.0 & 0.1 \\ 0.0 & 0.1 & 0.7 & 0.1 & 0.3 \\ 0.0 & 0.6 & 0.1 & 0.8 & 0.2 \\ 0.0 & 0.4 & 0.5 & 0.6 & 0.0 \end{Bmatrix} = \begin{bmatrix} 0.0000 & 0.2417 & 0.3151 & 0.2716 & 0.1716 \end{bmatrix}$$

$$B_{post-recovery\ society} = \begin{bmatrix} 0.4105 & 0.2551 & 0.1183 & 0.2161 \end{bmatrix} \cdot$$
$$\begin{Bmatrix} 0.3 & 0.2 & 0.2 & 0.1 & 0.1 \\ 0.1 & 0.0 & 0.3 & 0.4 & 0.4 \\ 0.5 & 0.6 & 0.4 & 0.2 & 0.1 \\ 0.5 & 0.7 & 0.7 & 0.2 & 0.2 \end{Bmatrix} = \begin{bmatrix} 0.1421 & 0.3024 & 0.1716 & 0.2158 & 0.1681 \end{bmatrix}$$

(4) Overall benefits

$$B_{pre-recovery\ overall} = \begin{bmatrix} 0.5102 & 0.3151 & 0.1788 \end{bmatrix} \cdot$$
$$\begin{bmatrix} 0.0000 & 0.1343 & 0.2581 & 0.3601 & 0.0715 \\ 0.0000 & 0.1513 & 0.3106 & 0.4031 & 0.1350 \\ 0.0000 & 0.2417 & 0.3151 & 0.2716 & 0.1716 \end{bmatrix} = \begin{bmatrix} 0.0000 & 0.1531 & 0.4062 & 0.3127 & 0.1290 \end{bmatrix}$$

$$B_{post-recovery\ overall} = \begin{bmatrix} 0.5102 & 0.3151 & 0.1788 \end{bmatrix} \cdot$$
$$\begin{bmatrix} 0.1120 & 0.2043 & 0.3521 & 0.1613 & 0.1703 \\ 0.2903 & 0.1572 & 0.2011 & 0.2091 & 0.1423 \\ 0.1421 & 0.3024 & 0.1716 & 0.2158 & 0.1681 \end{bmatrix} = \begin{bmatrix} 0.2061 & 0.1725 & 0.3019 & 0.1916 & 0.1279 \end{bmatrix}$$

The results of the real response evaluation directly reflected the ecological, economic, and social benefits, which are crucial for reflecting the comprehensive benefits before and after land reclamation. Based on the FCE, the index layer was evaluated, and the calculation results of different levels were obtained. Then, the difference value of each level of the classification matrix was set to 20. Using the grading standard of the evaluation system, the evaluation standard score matrix was established:

$$M = (f1,f2,f3,f4,f5)^T = (20,40,60,80,100)^T$$

(1) Ecological benefits

$$Z_{pre-recovery\ ecology} = B_{pre-recovery\ ecology} \times M =$$
$$\begin{bmatrix} 0.0000 & 0.1343 & 0.2581 & 0.3601 & 0.0715 \end{bmatrix} \times \begin{bmatrix} 100 \\ 80 \\ 60 \\ 40 \\ 20 \end{bmatrix} = 50.15$$

$$Z_{post-recovery\ ecology} = B_{post-recovery\ ecology} \times M =$$
$$\begin{bmatrix} 0.1120 & 0.2043 & 0.3521 & 0.1613 & 0.1703 \end{bmatrix} \times \begin{bmatrix} 100 \\ 80 \\ 60 \\ 40 \\ 20 \end{bmatrix} = 78.23$$

(2) Economic benefits

$$Z_{pre-recovery\ economy} = B_{pre-recovery\ economy} \times M =$$

$$\begin{bmatrix} 0.0000 & 0.1513 & 0.3106 & 0.4031 & 0.1350 \end{bmatrix} \times \begin{bmatrix} 100 \\ 80 \\ 60 \\ 40 \\ 20 \end{bmatrix} = 49.03$$

$$Z_{post-recovery\ economy} = B_{post-recovery\ economy} \times M =$$

$$\begin{bmatrix} 0.2903 & 0.1572 & 0.2011 & 0.2091 & 0.1423 \end{bmatrix} \times \begin{bmatrix} 100 \\ 80 \\ 60 \\ 40 \\ 20 \end{bmatrix} = 71.25$$

(3) Social benefits

$$Z_{pre-recovery\ society} = B_{pre-recovery\ society} \times M =$$

$$\begin{bmatrix} 0.0000 & 0.2417 & 0.3151 & 0.2716 & 0.1716 \end{bmatrix} \times \begin{bmatrix} 100 \\ 80 \\ 60 \\ 40 \\ 20 \end{bmatrix} = 53.41$$

$$Z_{post-recovery\ society} = B_{post-recovery\ society} \times M =$$

$$\begin{bmatrix} 0.1421 & 0.3024 & 0.1716 & 0.2158 & 0.1681 \end{bmatrix} \times \begin{bmatrix} 100 \\ 80 \\ 60 \\ 40 \\ 20 \end{bmatrix} = 70.51$$

(4) Overall benefits

$$Z_{pre-recovery\ overall} = B_{pre-recovery\ overall} \times M =$$

$$\begin{bmatrix} 0.0000 & 0.1531 & 0.4062 & 0.3127 & 0.1290 \end{bmatrix} \times \begin{bmatrix} 100 \\ 80 \\ 60 \\ 40 \\ 20 \end{bmatrix} = 55.28$$

$$Z_{post-recovery\ overall} = B_{post-recovery\ overall} \times M =$$

$$\begin{bmatrix} 0.2061 & 0.1725 & 0.3019 & 0.1916 & 0.1279 \end{bmatrix} \times \begin{bmatrix} 100 \\ 80 \\ 60 \\ 40 \\ 20 \end{bmatrix} = 78.16$$

We obtained the merit evaluation scores of the land reclamation of Hanjiawan coal mine in Shendong mining area based on the calculated results mentioned above and the grade evaluation criteria listed in Table 6 (results summarized in Table 10).

The quantitative results of various indicators present that the large-scale exploitation of coal resources led to relatively poor ecological, economic, social, and overall benefits. By implementing

land reclamation, the benefit scores of various indicators in mining areas substantially increased, with a growth rate of 32–56%. The ecological benefits increased significantly, and all indicators were improved after land reclamation.

**Table 10.** Evaluation value of the land reclamation benefits in Hanjiawan coal mine.

| Quantitative Indicators | Ecological Benefits | Economic Benefits | Social Benefits | Overall Benefits |
| --- | --- | --- | --- | --- |
| Before reclamation | 50.15 | 49.03 | 53.41 | 55.28 |
| Evaluation standard | Average | Average | Average | Average |
| After reclamation | 78.23 | 71.25 | 70.51 | 78.16 |
| Evaluation standard | Good | Good | Good | Good |
| Benefits growth value | 28.08 | 22.22 | 17.10 | 22.88 |

## 6. Results and Discussion

Based on the evaluation of the main evaluation indicators, the mining area of ecological benefits (B1), economic benefits (B2), and social benefits (B3) were all good. Thus, the results were discussed in the following three aspects.

### 6.1. Ecological Benefits

After the implementation of the reclamation and control project, the ecological environment benefits of the mining area will be increased, and the forest and grass coverage in the area will increase manifold, effectively enhancing the regional ecological environment, increasing the aerobic content, and purifying the air in the mining area, as well as avoiding the loss of soil nutrients and promoting the virtuous cycle of ecological environment. In addition, disturbed or destroyed vegetation and landforms are basically controlled, effectively decreasing soil and water loss, basically controlling soil and water loss, improving land quality, and increasing soil fertility. Through land reclamation, the subsidence land and saline-alkali land in the mining area can be turned into an oasis, and the desert land can be reclaimed, which could effectively expand the cultivated land area. Furthermore, the ecological landscape and environmental benefits attained a massive change from negative benefits to positive benefits.

### 6.2. Economic Benefits

As land reclamation in the mining area is a type of economic investment activity, it is economically feasible. Based on the reclamation planning and design of the direction of land use after reclamation, within a certain time range, the investment cost of reclamation can be recovered, and the direct economic benefits can be brought to mining area through land reclamation. Assumedly, the recoverable residual coal in the mining area after land reclamation is nearly 52 million tons. The calculation standard of the cost is as follows: the management cost is 8.00 yuan/ton, the laboring cost is 6.00 yuan/ton, the machinery cost is 11.00 yuan/ton, and the tax is 61.20 yuan/ton. The sale price of each ton of coal is 528 yuan, and the profit is 22,973.6 million yuan. Furthermore, the economic benefit of land reclamation in the mining area is very considerable. After the industrialization, its economic benefit potential will be higher, and it will also effectively promote the effective development of reclamation-related industries.

### 6.3. Social Benefits

Land reclamation and management in mining areas not only has great significance for the green and safe production of mining companies themselves but also plays a positive role in promoting the ecological, social, and economic harmonious and sustainable development of mining areas. In addition, land reclamation is conducive to the safety and green production of mining enterprises, so that mining enterprises can attain the highest social and economic benefits. Meanwhile, it can ensure the sustainable use of land resources in the mining area, ecological environment, inclusive economic growth, and social harmony. Moreover, stable, orderly and healthy living, and production of residents played a

part in a green ecological landscape, while promoting the coordinated and sustainable development of local industry, agriculture, forestry, and animal husbandry.

## 7. Conclusions

Land reclamation in Shendong mining area plays an important role in the economic development of this region. Using the monitoring data of land reclamation in the mining area, combined with the status quo of land reclamation, a land reclamation evaluation system was established for Hanjiawan coal mine in Shendong mining area, and the following conclusions were obtained:

By implementing land reclamation, the land reclamation area in Shendong mining area increased by 63%, and the land types markedly improved. The land reclamation decreased the amount of damaged lands, such as subsidence land, accumulation land, and mining land, and also increased the agricultural land, construction land, and forestry land. Based on the suitability of land reclamation and feasibility analysis, it was deduced that the land Shendong mining area was highly suitable for land reclamation, and that the post-reclamation conditions would be stable.

The FCE comprehensive evaluation not only calculated the ecological, economic, and social benefits before and after land reclamation, but also helped obtain the grade standard of benefit evaluation of the mining region after land reclamation. This study suggests that, by implementing reclamation, Hanjiawan coal mine's benefit score could be significantly increased at a growth rate as high as 56%.

Land reclamation offers good ecological, economic, and social benefits and is highly crucial to safe and sustainable agricultural production, as well as the sustainable and healthy development of industrial mining enterprises. Furthermore, land reclamation plays an active role in improving the habitable environment of residents and the ecological landscape, as well as contributing to the economic development of the mining region.

**Author Contributions:** Conceptualization, C.M.; experimental Design, C.M., X.Y. and D.Z.; validation, C.M. and X.Y.; theoretical analysis, C.M. and X.Y.; data curation, C.M. and D.Z.; supervision, X.Y.; writing—original draft preparation, C.M.; writing—review and editing, X.Y. and C.M.; supervision, X.Y.; project administration, X.Y.; funding acquisition, X.Y. All authors have read and agreed to the published version of the manuscript.

**Funding:** This research was funded by the National Natural Science Foundation of China, grant number No. 51874230.

**Acknowledgments:** We thank the National Natural Science Foundation of China for its support of this study. We thank the academic editors and anonymous reviewers for their kind suggestions and valuable comments.

**Conflicts of Interest:** The authors declare no conflict of interest.

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
