# Peer review of "Assessment of Land Reclamation Benefits in Mining Areas Using Fuzzy Comprehensive Evaluation"

_sustainability, doi:10.3390/su12052015_

Round 1

Reviewer 1 Report

Please find more details in the attached file.

Author Response

Manuscript ID: sustainability-700663

Type: Article

Title: Study on Evaluation of Land Reclamation Benefit in Mining Area Based on Analytic Hierarchy Process

Correspondence Author: Chi Mu, Email: muc@stu.xust.edu.cn

Thank you for your letter and for the reviewer's comments concerning our manuscript entitled "Study on Evaluation of Land Reclamation Benefit in Mining Area Based on Analytic Hierarchy Process", Those comments are all valuable and very helpful for revising and improving our paper. We have studied comments carefully and have made correction which we hope meet with approval. Revised portion are marked in the paper. The responds to the reviewer's comments are as following:

Response to Reviewer 1 Comments

Point 1: L45 - Domestic scholars mean Chinese scholars? Any worldwide scholars working on similar topics, i.e. from Australian/Brazil mining industry and research groups? 

Response 1: Domestic scholars mean Chinese scholars. Considering the reviewer's suggestion that we have clearly marked the country information of the research scholars in the manuscript.

Point 2: L52 - Delete the space before comma.

Response 2: The space before the comma has been deleted.

Point 3: L51-53 - Please reorganise sentence.

Response 3: We have re-written this sentence according to the Reviewer’s suggestion.

Point 4: L56 - Please identify these terms: “VIKOR”, “GIS”, “RS”, “ANN”, “ESRI”.

Response 4: It is really true as reviewer suggested that these terms "GIS", "RS", "ANN", "ESRI" have been identified. VIKOR method is one of the frequently used multi-attribute decision making, which provides a compromise solution with maximum group utility.

Point 5: L57 - Please change the full stop to comma.

Response 5: It has been revised in the manuscript.

Point 6: L67-69 - Please reorganise the sentence.

Response 6: We have re-written this sentence according to the Reviewer’s suggestion.

Point 7: L45-75 - The authors listed quite a few current models of mining land reclamation. However, their differences (i.e., consideration parameters, the characterisation method in obtaining these parameters, the way to conduct the model, etc.) and gaps haven’t been clearly elaborated.

Response 7: We have revised it according to the reviewer's comment, and the evaluation methods (feature, advantage and disadvantage) are compared in the new manuscript (see revised manuscript Table 1).

Point 8: L127 - The authors seek to evaluate the land reclamation benefits and list the consideration parameters in Table 2. However, something bothers me is that this model hasn’t take the reclamation method and its associated (labouring, machinery, amendments) costs into consideration. For example, the high cost input reclamation may result in better land reclamation benefits in using current model, however, the overall benefits assessment may not as high as it looks if all the initial reclamation input cost been considered (deducted).

Response 8: Considering the reviewer’s suggestion, we have considered (labouring, machinery, amendments) costs of land reclamation in the revised manuscript (see revised manuscript Table 7). We have discussed economic benefits after land reclamation in subchapter 6.2 and considered land reclamation methods in subchapter 4.2.

Point 9: L134 - Please be careful of the marks, the second and third comma in equation 1 is different. Please revise it elsewhere applies.

Response 9: It has been revised in the manuscript and carefully checked elsewhere.

Point 10: L192 - What’s “the thickness of soil layer”? Deformed soil layer? Collapsed soil layer? Retained soil layer? This requires further clarification.

Response 10: We are very sorry for our incorrect writing of “thickness of soil layer”. The correct writing is "thickness of surface soil". The thickness of surface soil is the thickness from the ground surface to the lower clay.

Point 11: L181-192 - An interesting question is that how the authors conduct the real time investigation, via satellite images, human visual checking, or other specified methods? How is the accuracy?

Response 11: We set up a surface mobile observation station in the mining area, and use a total station to stake out the measurement points based on the principle of polar coordinates. During the setting out, we can make appropriate adjustments according to the actual situation, wait for the observation points to stabilize, and then observe the data. This method has high accuracy and can meet the requirements of surface monitoring in mining areas.

Point 12: L202 - Space between figure and unit.

Response 12: It has been revised in the manuscript.

Point 13: L239 - My thoughts would be the evaluation index should be a variable rather than a fixed value, which is depending on each parameter and their interactions.

Response 13: As reviewer suggested that we separately count the occurrence frequency of individual (ecological, economic, and social) benefit evaluation indicators in relevant references. First, according to the evaluation indicators with higher frequency, they are arranged in descending order. For example, there are 36 evaluation indicators that contribute to economic benefits (Land use efficiency in mining area, input-output ratio, per capita income of residents et al). Second, merge similar items, select common indicators, discard individual indicators, analyze the connotation of land reclamation benefits in mining areas, and add evaluation indicators that meet the actual conditions of mining areas. Finally, the evaluation index system of land reclamation benefit in mining area is established.

Special thanks to you for your good comments.                       

Reviewer 2 Report

The article shows interesting and modern aproach to a problem that is described, however there are some understatements that need to be clarified.

Some general comments:

Line 51: "RS" abbreviation is not common known, please explain it before first use.

2. Line 59: "ANN" abbreviation could be not common known, please explain it before first use.

3. A map showing the location of the study area should be attached.

4. Please consider moving chapter 3 - "Feasibility analysis of land reclamation in the mining area" after 1.Introduction and before 2.Evaluation method of land reclamation in mining area. It describes the situation and the evaluation problem being tried to be solved by the methods used.

5. Subchapter 2.4 should definitelly be placed before 2.3. According to AHP algorithm we should first define the structure of hierarchy before the main analysis is done.

Some specific comments:

6. Shouldn't there be arrows from "Analysis the land reclamation impact factor" to "Quantify the evaluation factor and calculate its weight value" and continuing to "Test the matrix consistency" in figure 1? Do those three steps happen simultaneously? Don't the next one follows from the previous step?

7. The content of the last column of Table 2 should be increased by a description of the data source.

8. Is Table 3 correct?

9. How were the fuzzy evaluation matrices before and after land reclamation established? It isn't described at all, there are only some general statements included in the text.

10. Shouldn't there be 6 rows in Rpre-recovery ecology and Rpost-recovery ecology matrices? Are the matrices correct?

11. Line 101: What is the "Factor set U" and how was it partitioned into a subset of s factors according to some defined property c? What is "c"? It seems that the references mentioned before and after that part of text do not describe that problem. How are this references revelant to this problem?

Author Response

Manuscript ID: sustainability-700663

Type: Article

Title: Study on Evaluation of Land Reclamation Benefit in Mining Area Based on Analytic Hierarchy Process

Correspondence Author: Chi Mu, Email: muc@stu.xust.edu.cn

Thank you for your letter and for the reviewer's comments concerning our manuscript entitled "Study on Evaluation of Land Reclamation Benefit in Mining Area Based on Analytic Hierarchy Process", Those comments are all valuable and very helpful for revising and improving our paper. We have studied comments carefully and have made correction which we hope meet with approval. Revised portion are marked in the paper. The responds to the reviewer's comments are as following:

Response to Reviewer 2 Comments

Point 1: "RS" abbreviation is not common known, please explain it before first use.

Response 1: It is really true as reviewer suggested that the term "RS" has been identified in the manuscript.

Point 2: "ANN" abbreviation could be not common known, please explain it before first use.

Response 2: It has been explained it before first use in the manuscript.

Point 3: A map showing the location of the study area should be attached.

Response 3: As reviewer suggested that a map of the location of the study area is shown in the manuscript (see revised manuscript Figure 2).

Figure 2. Location of the study area

Point 4: Please consider moving chapter 3 - "Feasibility analysis of land reclamation in the mining area" after 1.Introduction and before 2.Evaluation method of land reclamation in mining area. It describes the situation and the evaluation problem being tried to be solved by the methods used.

Response 4: Considering the reviewer's suggestion that we have moved chapter 3 - "Feasibility analysis of land reclamation in the mining area" after 1.Introduction and before 2.Evaluation method of land reclamation in mining area.

Point 5: Subchapter 2.4 should definitelly be placed before 2.3. According to AHP algorithm we should first define the structure of hierarchy before the main analysis is done.

Response 5: It is really true as reviewer suggested that we should first define the structure of hierarchy before the main analysis is done. So we have placed subchapter 2.4 before 2.3 (see revised manuscript subchapter 4.4 and subchapter 4.5).

Point 6: Shouldn't there be arrows from "Analysis the land reclamation impact factor" to "Quantify the evaluation factor and calculate its weight value" and continuing to "Test the matrix consistency" in figure 1? Do those three steps happen simultaneously? Don't the next one follows from the previous step?

Response 6: We are very sorry for our negligence of incorrect step drawing. We have redrawn the steps of comprehensive evaluation (see revised manuscript Figure 5).

Figure 5. Steps of comprehensive evaluation

Point 7: The content of the last column of Table 2 should be increased by a description of the data source.

Response 7: Considering the reviewer's suggestion, we have added a description of the data source in the last column and renamed it Table 7.

Point 8: Is Table 3 correct?

Response 8: Table 3 is incorrect. We have modified it and renamed it Table 8.

Table 8. Judgment matrix randomness index

Order

1

2

3

4

5

6

7

8

9

10

11

12

13

14

RI

0

0

0.52

0.89

1.12

1.24

1.36

1.41

1.46

1.49

1.52

1.54

1.56

1.58

Point 9: How were the fuzzy evaluation matrices before and after land reclamation established? It isn't described at all, there are only some general statements included in the text.

Response 9: We quantify the standard layer based on basic data and expert scores (an expert group of industry experts), combined with the expert's judgment of the evaluation results before and after land reclamation, we established a fuzzy evaluation matrix (see revised manuscript subchapter 5.1).

Point 10: Shouldn't there be 6 rows in Rpre-recovery ecology and Rpost-recovery ecology matrices? Are the matrices correct?

Response 10: We are very sorry for our incorrect writing of matrix. There are 6 rows in Rpre-recovery ecology and Rpost-recovery ecology matrices, and it has been revised in the manuscript.

Point 11: What is the "Factor set U" and how was it partitioned into a subset of s factors according to some defined property c? What is "c"? It seems that the references mentioned before and after that part of text do not describe that problem. How are this references revelant to this problem?

Response 11: We are very sorry that this part of the content is not explained clearly in the manuscript. In order to make the reviewers understand the process of establishing the comprehensive evaluation model, we have rewritten this part (see revised manuscript subchapter 4.1).

Special thanks to you for your good comments.

Reviewer 3 Report

Dear Editor,

Its well-written and possible to publish after incorporating the comments in attached file.

Author Response

Manuscript ID: sustainability-700663

Type: Article

Title: Study on Evaluation of Land Reclamation Benefit in Mining Area Based on Analytic Hierarchy Process

Correspondence Author: Chi Mu, Email: muc@stu.xust.edu.cn

Thank you for your letter and for the reviewer's comments concerning our manuscript entitled "Study on Evaluation of Land Reclamation Benefit in Mining Area Based on Analytic Hierarchy Process", Those comments are all valuable and very helpful for revising and improving our paper. We have studied comments carefully and have made correction which we hope meet with approval. Revised portion are marked in the paper. The responds to the reviewer's comments are as following:

Response to Reviewer 3 Comments

Point 1: First, however, the study seems to be a part of a large study on evaluation of Land Reclamation Benefit Area based on AHP, and it is quite confusing and not clear that AHP is the only method used. For instance, the abstract tells that furthermore, a hierarchical analysis was used to perform a land reclamation evaluation using an analytical hierarchy process. I therefore suggest to re-write the topics: replace based on or USING AHP. Second, the introduction part provides the domestic scholars but there is still not clear message of region or country of study area. The study shows that land reclamation significantly the land type within the mining region whereas AHP is especially suited for cases for the evaluation but the result of the study also claims that there is also improved both agriculture and construction land and evaluation criteria after reclamation were good. Besides, my experience tells that it depends the number of criteria and indicators and how AHP were used and accommodate the views of different stakeholders including the conflicting and compromising elements. The abstract tells the proposed method (?) could be used to accurately evaluate various benefits obtain after land reclamation. In my experiences, AHP is based on multi-criteria analysis to identify the best alternative options and demonstrate the tradeoff and sensitive analysis among given problems or solutions. Discussion talks about importance of sensitivity analysis, but I could not see what was the sensitivity analysis the authors were talking about. Discussion was talking about how this sort of process does not produce how coordinate sustainable ecological and economic development and to find the optimal solution as a rational-based decisions. So, I want to see the authors can provide what such an optimal solution in a given problem.

Response 1: We're sorry that we didn't give a clear description of the method used. It is really true as reviewer suggested that AHP is not the only method used. We also used the fuzzy mathematics method in the evaluation process. Fuzzy comprehensive evaluation (FCE) is the combination of AHP and fuzzy mathematics. Considering the reviewer's suggestion that we have re-written the topics: Assessment of Land Reclamation Benefit in Mining Area Using Fuzzy Comprehensive Evaluation. To let reviewers know how to use AHP and build a comprehensive evaluation model, we have rewritten this part (see revised manuscript subchapter 4.1). Considering the reviewer's suggestion that we have clearly marked the country information of the research scholars in the manuscript. Domestic scholars mean Chinese scholars. AHP effectively solves multilevel and multi-objective decision-making problems in large systems and has the characteristics of high logic, flexibility, and simplicity. The land reclamation project in the mining area involves many types of factors and contains complicated links. There could be many methods to choose from during reclamation. We can sort out several representative methods, apply AHP for selection analysis, and obtain the best reclamation effect with as few reclamation methods as possible, as well as obtain good ecological, economic, and social benefits.

Point 2: The overall titles tells the study was evaluated based on AHP but its given using AHP and others traditional and experts methods. As earlier mention, AHP methods can/could be used by using different mix methods in order to assess and evaluate their elements as so called Criteria and Indicators.

Response 2: It is really true as reviewer suggested that using different mix methods in order to assess and evaluate their elements. We also use the fuzzy mathematics method in the evaluation process. Fuzzy comprehensive evaluation (FCE) is the combination of AHP and fuzzy mathematics.

Point 3: Analytic Hierarchy Process or Analytical Hierarchy Process?

Response 3: Analytic Hierarchy Process

Point 4: Where are the ideas mentioned in the abstract deal with in the study? The case study region or states are not given.

Response 4: According to the ideas mentioned in the abstract, we evaluated the feasibility of land reclamation in the mining area based on the ecological landscape effect, the area of agricultural land reclamation, and the residents' satisfaction. Through the consistency check of the judgment matrix, an evaluation model of land reclamation in the mining area was established. According to the evaluation criteria, the expert scoring method is adopted, and the ecological, economic and social benefits before and after land reclamation in the mining area are also calculated. As reviewer suggested that a map of the location of the study area is shown in the manuscript (see revised manuscript Figure 2).

Figure 2. Location of the study area

Point 5: Could you mention somehow what are the real problems and different alternative options including number of evaluation criteria?

Response 5: The real problem is that the over exploitation of mineral resources damages the ecological, economic and social benefits of the mining area, so it is necessary to calculate the ecological, economic and social benefits before and after the land reclamation of the mining area. As reviewer suggested that we separately count the occurrence frequency of individual (ecological, economic, and social) benefit evaluation indicators in relevant references. First, according to the evaluation indicators with higher frequency, they are arranged in descending order. For example, there are 36 evaluation indicators that contribute to economic benefits (Land use efficiency in mining area, input-output ratio, per capita income of residents et al). Second, merge similar items, select common indicators, discard individual indicators, analyze the connotation of land reclamation benefits in mining areas, and add evaluation indicators that meet the actual conditions of mining areas. Finally, the evaluation index system of land reclamation benefit in mining area is established.

Point 6: Like In line 50-52, Line spacing, 63,…grammar, spelling and space to be corrected. Like consistency in table and other formats e.g. Table 3 table. Full stop at the end

Response 6: We are very sorry for our incorrect writing of grammar, spelling and space. According to the manuscript template, we corrected the format and grammar of the manuscript and carefully checked elsewhere.

Point 7 Like abstract, there is mentioned evaluation index, non-linear evaluation theory and other key terms but not clearly described in introductory parts. Why such theory and index are essential for the evaluation proposes…

Response 7: The evaluation index is the foundation of the establishment of the comprehensive evaluation system of the mining area. The change of the evaluation index not only has a great contribution to the scope of the mining area, but also to the ecology, economy and society of the mining area. As time goes on, its effect will become more and more obvious. Considering the use of fuzzy comprehensive evaluation (FCE) method, the non-linear evaluation theory is removed from the manuscript.

Point 8: There is given some example and clear what the earlier studies have been about and what is the new contribution of this study related to the earlier one. All concepts used should be properly defined and the discussion should be about the study at hand.

Response 8: It is really true as reviewer suggested that all concepts used should be properly defined. We have defined all the concepts in the manuscript before first use.

Point 9: I will see how the proposed methods and tools have been applied? Are these the traditional ones mentioned to develop the plan and optimal solutions? If yes, then the reader should know what and how sorts of tools were employ.

Response 9: We are very sorry that this part of the content is not explained clearly in the manuscript. In order for the reviewers to understand how the methods and tools are applied, we have described the process of establishing a comprehensive evaluation model by using AHP (see revised manuscript subchapter 4.1). A fuzzy evaluation matrix was established by using fuzzy mathematics, combined with the expert's judgment of the evaluation results before and after land reclamation, we established a fuzzy evaluation matrix (see revised manuscript subchapter 5.1).

Point 10: There is given several evaluation methods but I don’t see any connection whatsoever. Is this process what the discussion is about? Are these explained in some earlier study?

Response 10: Based on the evaluation results of land reclamation in mining areas, experts and scholars commonly use the following evaluation methods: analytic hierarchy process (AHP); fuzzy comprehensive evaluation (FCE); principal component analysis (PCA); fault tree analysis (FTA); and logical framework approach (LFA). All evaluation methods have various differences. Table 1 compares the common evaluation methods. We also used the fuzzy mathematics method in the evaluation process. FCE is the combination of AHP and fuzzy mathematics.

Table 1. Comparison of evaluation methods

Method

Feature

Advantage

Disadvantage

Analytic hierarchy process

(AHP)

Combination of qualitative and quantitative analysis

Systematically treat research objects as a system and make decisions in a comprehensive way of thinking.

Cannot provide new options for decision-making.

Fuzzy comprehensive evaluation

(FCE)

Combination of AHP and fuzzy mathematics

The problem of quantification of a large number of uncertain factors in the evaluation is well solved.

It is more subjective when determining fuzzy relation matrix and factor weight.

Principal component analysis

(PCA)

Determine the number of principal components through appropriate mathematical changes

Can eliminate the mutual influence between evaluation indicators and reduce calculation workload

The raw data requirements are complete and the evaluation process is not flexible.

Fault tree analysis

(FTA)

Create a list of key events for different importance measurements.

Use graphics to clearly show the process and results of the evaluation object

It is no way to accurately calculate system failure probability and reliability.

Logical framework approach

(LFA)

Use simple block diagrams to analyze complex relationships

Clarify the cause and effect in the project

Greatly affected by important assumptions

Point 11: It seems to me that ordinal data was collected, the answers were aggregated somehow (not possible to understand how), and the aggregated result (ordinal data still) was somehow transformed to ratio-scale (impossible to know how), and finally geometric means were taken. I suspect the results very much depend on how this was carried out. For instance, the analysis of consistency is meaningless, as it is the consistency of this transformation that is analyzed, not the consistency of the answers. A sensitivity analysis on this would be sorely needed. If the readers cannot see that the proposed method is not distorting the results, the method is simply not acceptable. It is an ad hoc method at best.

Response 11: This study decomposes the elements related to decision-making into levels such as target, criteria, and index. Based on this, it makes a qualitative and quantitative analysis of decision-making methods. Its usage is to construct a judgment matrix and find its maximum eigenvalue. When determining the order of a matrix, it is often difficult to construct a matrix that satisfies the consistency. However, the judgment matrix deviates from the consistency condition by another degree. To this end, it is necessary to judge whether the judgment matrix is acceptable. In conclusion, the analysis of consistency is necessary in land reclamation benefit evaluation.

Point 12: In conclusion, its given the suitably of land reclamation and feasibility analysis, its confuse how this information was used in the evaluation process. Please be explicit?

Response 12: The suitably of land reclamation and feasibility analysis is the basis of comprehensive benefit evaluation. Only by analyzing the ways of land destruction and factors can we evaluate the ecological, economic and social benefits of land reclamation in mining areas. According to the suitably of land reclamation and feasibility analysis, the evaluation results are more in line with the actual situation of the mining area.

Point 13: There is also given sustainable and healthy development of industrial mining enterprises would be essential but It was never mentioned in results nor methods.

Response 13: Considering the reviewer's suggestion that we have given the sustainable and healthy development of industrial mining enterprises in the "Results and Discussion" section of the manuscript (see revised manuscript Chapter 6).

Special thanks to you for your good comments.

Round 2

Reviewer 1 Report

The previous questions have been answered and most are addressed. 

The authors should focus on English writing to improve the readability of this paper. 

Author Response

Manuscript ID: sustainability-700663

Type: Article

Title: Assessment of Land Reclamation Benefit in Mining Area Using Fuzzy Comprehensive Evaluation

Correspondence Author: Chi Mu, Email: muc@stu.xust.edu.cn

Thank you for your letter and for the reviewer's comments concerning our manuscript entitled "Assessment of Land Reclamation Benefit in Mining Area Using Fuzzy Comprehensive Evaluation ". Those comments are all valuable and very helpful for revising and improving our paper. We have studied comments carefully and have made correction which we hope meet with approval. Revised portion are marked in the paper. The responds to the reviewer's comments are as following:

Response to Reviewer 1 Comments

Point: The previous questions have been answered and most are addressed. The authors should focus on English writing to improve the readability of this paper.

Response: It is really true as reviewer suggested that we have extensively revised English writing and checked the manuscript using a professional English editing service.

Special thanks to you for your good comments.                       

Reviewer 2 Report

I have no doubt that almost all my comments have been taken into account and thank you for understanding them.

  1. However, I have again comments on Table 7. Previously, I have suggested that the authors should add the source from which they took the data for analysis. It was more about determining the origin of the data (digital maps?, statistical data from government databases? Etc) rather than the way they were calculated.
  2. In addition, after rethinking, I think that Figure 7 is not needed. It does not contain additional information, it only replicates the ones contained in Table 10.
  3. And one last small note - please explain "AHP" abbreviation in line 49 first.

I leave the matter of the significance of the above comments to the editor's decision.

Author Response

Manuscript ID: sustainability-700663

Type: Article

Title: Assessment of Land Reclamation Benefit in Mining Area Using Fuzzy Comprehensive Evaluation

Correspondence Author: Chi Mu, Email: muc@stu.xust.edu.cn

Thank you for your letter and for the reviewer's comments concerning our manuscript entitled "Assessment of Land Reclamation Benefit in Mining Area Using Fuzzy Comprehensive Evaluation ". Those comments are all valuable and very helpful for revising and improving our paper. We have studied comments carefully and have made correction which we hope meet with approval. Revised portion are marked in the paper. The responds to the reviewer's comments are as following:

Response to Reviewer 2 Comments

Point 1: However, I have again comments on Table 7. Previously, I have suggested that the authors should add the source from which they took the data for analysis. It was more about determining the origin of the data (digital maps?, statistical data from government databases? Etc) rather than the way they were calculated.

Response 1: It is really true as reviewer suggested that we have added the origin of the data in Table 7.

Point 2: In addition, after rethinking, I think that Figure 7 is not needed. It does not contain additional information, it only replicates the ones contained in Table 10.

Response 2: Considering the reviewer's suggestion that Figure 7 has been deleted.

Point 3: And one last small note - please explain "AHP" abbreviation in line 49 first.

Response 3: It has been explained "AHP" abbreviation in the revised manuscript.

Special thanks to you for your good comments.                  
